# Integrating traditional medicine into antimicrobial resistance education: Community-centric preparedness in Zimbabwe

Martin Mickelsson[1,2]*, Tungamirirai Simbini[3]

1 Department of Earth Sciences, University of Gothenburg, Gothenburg, Sweden, 2 Department of Women's and Children's Health, Uppsala University, Uppsala, Sweden, 3 Department of Biomedical Informatics and Biomedical Engineering, University of Zimbabwe, Harare, Zimbabwe

* martin.mickelsson@gu.se

## Abstract

Antimicrobial resistance (AMR) constitutes a growing risk to global health, particularly in low- and middle-income countries (LMICs) like Zimbabwe. AMR education has often focused on prevention, with limited emphasis on preparedness, especially in community-based approaches. Education and practice frequently overlook how preparedness can limit the spread of resistance once it has emerged. Traditional medicine, a key component of primary health care in Zimbabwe, may offer culturally grounded approaches to AMR preparedness. This study explores the role of traditional medicine in AMR preparedness in Zimbabwe. Using a participatory research workshop method, it included 25 health practitioners (medical doctors, nurses, and pharmacists) from two major teaching hospitals in Harare. The workshops explored practitioners' perceptions of traditional medicine in relation to AMR education and preparedness. Generated research data included audio recordings, which were analysed thematically. The participatory method enabled knowledge co-creation among practitioners during the discussions. Three themes emerged in the results: (1) traditional medicine is central to community healthcare access and early-stage treatment, especially in rural areas, reducing the need for antimicrobials; (2) it complements conventional medical practices by supporting health management and resilience, reducing antimicrobial dependency; and (3) traditional practitioners, embedded in communities, are key in disease surveillance through early outbreak detection. Recurring across these themes is the patient- and community-centric character of traditional medicine that supports AMR preparedness. This paper illustrates how traditional medicine is a crucial but often underutilized resource for preparedness in AMR education, offering community-based and culturally adapted approaches in Zimbabwe. Integrating traditional medicine into AMR education could strengthen health system resilience and public health outcomes, particularly in LMICs. AMR policy

**Data availability statement:** The data used in this study involve human subjects research participant data and are considered sensitive data. They are not publicly available due to restrictions outlined in a data use agreement. Access to the dataset is limited to authorized researchers and governed by confidentiality and privacy protections. For persons interested in obtaining these data, please contact: Cecil G. Sheps Center for Health Services Research The University of North Carolina at Chapel Hill CB# 7590 725 Martin Luther King Jr. Blvd. Chapel Hill, NC 27599-7590 contact@schsr.unc.edu P: 919-966-5011.

**Funding:** This work was supported by Vetenskapsrådet under Grant 2020-04567 to MM. The funders had no role in study design, data collection and analysis, decision to publish, or preparation of the manuscript.

**Competing interests:** The authors have declared that no competing interests exist.

efforts should promote integration with conventional practices to leverage traditional knowledge in building more resilient and effective approaches to resistance.

## Introduction

In response to antimicrobial resistance (AMR) as a global sustainability and health equity challenge with disproportionate impact on vulnerable communities, this study understands AMR education as a pathway for inclusive and culturally relevant health efforts [1–3]. AMR education, including in Zimbabwe, has traditionally prioritised the promotion of prevention, understood as efforts to limit inappropriate antimicrobial use and reduce infection risk, reduce the need for antimicrobials and thus avoid the emergence of resistance [4,5]. This includes antimicrobial stewardship, water, sanitation and hygiene (WASH) efforts as well as public awareness campaigns [6–8]. While prevention is essential, it is not sufficient for a comprehensive approach to AMR education.

Less attention has been given to preparedness, defined as the capacity to identify and manage the spread of resistant infections once they have emerged. Preparedness includes timely outbreak response, community engagement in containment, and the development of resilient health infrastructure. The focus on prevention in AMR education reflects a broader tendency in AMR responses to emphasise prescriber behaviour and individual patient responsibility, while de-emphasising upstream structural drivers of AMR. As seen in stewardship approaches, surveillance systems, protocols, and prescription management have been the main focus, with less attention given to poverty, housing, water and sanitation, and health system access and quality, especially in the rural-urban divide as critical factors shaping antimicrobial need and use [9,10]. While preventive educational efforts in communities are essential, they can create gaps in AMR education if framed around the misconception that resistance and resistant infections can be fully eliminated. As Blair et al. [11] and Laxminarayan et al. [12] argue, microbial resistance is an inevitable outcome of evolutionary processes, including spontaneous mutations, where microbes independently develop resistance genes, and horizontal gene transfer, in which microbes acquire resistance genes from others. Although antimicrobial overuse accelerates selection for resistance, these evolutionary mechanisms ensure that resistance will emerge even under strict antimicrobial stewardship. For this reason, AMR education must give equal weight to prevention and preparedness, prevention to slow the emergence of resistance, and preparedness to ensure effective responses once resistant infections occur. This imbalance is particularly problematic in low- and middle-income countries (LMICs) such as Zimbabwe, where chronic underfunding, shortages of trained personnel, and stark urban–rural divides exacerbate health inequalities and limit both preventive and responsive capacities [13–16]. In such contexts, focusing narrowly on prescriber practices or individual behaviours risks obscuring the systemic conditions that drive AMR, making it difficult to develop balanced approaches that integrate both prevention and preparedness. Without sufficient emphasis on both

dimensions, health practitioners and communities may lack the knowledge and practices necessary to manage resistant infections, leading to reactive rather than proactive outbreak responses when first-line antimicrobials fail [17]. These risks disproportionately affect vulnerable communities, which face higher exposure to infection due to inadequate socio-economic infrastructure and are often less able to access formal healthcare, including diagnostics, medicines, and treatment in the event of an outbreak [13,18].

Preparedness is thus distinct but complementary to prevention with the lack of clear distinctions between prevention and preparedness in AMR education having consequences for how resistance is addressed in policy and practice, particularly in low- and middle-income countries (LMICs) like Zimbabwe. Structural inequalities, limited resources and urban-rural divides exacerbate health vulnerabilities as capacity to proactively respond to resistant infections remains limited [13–16]. The World Health Organisation [19] in its report *Strengthening health emergency prevention, preparedness, response and resilience,* calls for equal emphasis on both prevention and preparedness in addressing global health challenges such as AMR.

Through AMR education that includes both prevention and preparedness, communities are better equipped to respond optimally to AMR in their communities. Furthermore, the emphasis on the formal health care system might result in prevention-centred AMR education leading to an underappreciation of the role of communities and limit opportunities to engage local knowledge systems and practices, such as traditional medicine [20]. AMR education should thus promote the prevention of resistance while simultaneously developing the ability of health practitioners and communities to proactively respond to resistant outbreaks by identifying and managing outbreaks collaboratively and locally, when prevention fails [21].

When considering these questions of prevention and preparedness in the national context of Zimbabwe, the Zimbabwean National Action Plan on AMR [22] acknowledges both but prioritises prevention. This is reflective of the WHO Global Action Plan on AMR [23], often preparedness limiting preparedness to a top-down process of policy implementation and enactment of crisis plans directed by national governments or the WHO without significant engagement of communities [22,23]. This illustrates a theme of linking preparedness to policy and government efforts. A more holistic and community-oriented approach to preparedness is needed that aligns with the definition by the WHO of health as complete physical, mental and social well-being [24]. This understanding also aligns with how WHO describes traditional medicine as part of global health systems:

> [...] the sum total of the knowledge, skills, and practices based on the theories, beliefs, and experiences indigenous to different cultures, whether explicable or not, used in the maintenance of health as well as in the prevention, diagnosis, improvement, or treatment of physical and mental illness [25, p. 44].

Traditional medicine, with a longstanding role in Zimbabwean community health, could provide a more patient-centric and community-oriented approach to address the preparedness gap in AMR education. Centring on community engagement, access and early identification of infections, traditional medicine is often the primary access to healthcare in rural and peri-urban areas. As such, traditional medicine and traditional practitioners are key for developing preparedness in areas where there is limited access to formal health care. Traditional practitioners are often the first point of contact for healthcare in communities, as they live within these communities and are embedded in local social structures, which fosters trust among community members. They thus become important as part of the preparedness for identifying, containing and managing resistant infection outbreaks and reducing the spread of resistant microbes and genes. In addition, their embeddedness within the community enables traditional practitioners to identify persistent health concerns, thereby contributing to resistant infection surveillance and broader preparedness efforts. Consequently, they are well-positioned to serve as key actors in strengthening local AMR preparedness through community education. If integrated well, traditional medicine can contribute to preparedness as a localised and proactive component of AMR education focused on health equity,

especially for vulnerable communities [26–28]. Despite acknowledging this potential of traditional medicine, research has seldom engaged with the absence of traditional medical practices from national AMR educational frameworks in Zimbabwe, limiting the extent and cultural relevance of national AMR education efforts [22]. This study addresses the identified gap by exploring how traditional medicine can be integrated in AMR education to support preparedness. The following two research questions guide the paper:

• What role does traditional medicine play in addressing AMR in Zimbabwe?

• How can traditional medical practice contribute to AMR preparedness?

## Materials and methods

### Ethics statement

The Medical Research Council of Zimbabwe (MRCZ/A/2920) approved this research. During the planning stage, the researchers sought permission from the clinical directors at Parirenyatwa Hospital and Sally Mugabe (Harare) Hospital to conduct workshops with health practitioners. The study followed the principles of informed consent and voluntary participation. The researchers provided participants with consent forms and information sheets at the start of the workshops. The information sheets explained the aims of the research, the workshop process, procedures for maintaining confidentiality, the types of data to be collected, and how the data would be managed. At the beginning of each workshop, participants had the opportunity to raise questions or concerns about the research or the workshop process. Afterward, they read the consent forms thoroughly and signed them. The researchers also encouraged participants to raise further questions during the workshops, addressing them as they arose. To protect confidentiality, the researchers assigned participants numerical codes and did not record their names during data collection. Throughout the study, the researchers adhered to the Global Code of Conduct for Research in Resource-Poor Settings [29].

### Research methods and materials

Two participatory research workshops with Zimbabwean health practitioners formed the basis of this study [30–32]. These were held separately at each of the teaching hospitals. Participatory workshops were selected as the primary method because they enable collective knowledge construction, facilitate the exchange of professional and community-grounded perspectives, and are well suited to exploring complex health challenges such as antimicrobial resistance (AMR), which cut across clinical, social, and cultural domains. This method has been previously utilised in research in Southern Africa to engage in knowledge co-creation around complex sustainability challenges together with practitioners. Results highlighted how participatory workshops can enable practitioners to contextualise sustainability challenges in relation to their lived experiences in developing adaptive approaches to addressing sustainability challenges [33,34], emphasising the method's contextual relevance. In developing and preparing the participatory workshops for this study we drew on this previous research in the region and were guided by our research aim. Adapting this participatory methodology through the shift to health and the addition of structured inputs on key AMR themes as stimulus for discussion, our approach actively centred local health practitioners as co-creators of knowledge rather than passive subjects, ensuring their voices directly shaped the research insights.

The workshops explored the role of traditional medicine in AMR education, prevention, and preparedness. Participants first received structured input on key themes, including global and local AMR trends, pathways of resistance development, WHO strategies on antimicrobial stewardship, and the importance of integrating prevention and preparedness into community health practice. This provided a shared foundation for discussion. Participants then contributed through facilitated group activities, in which they shared practical experiences of AMR challenges in their clinical work, described traditional medicine practices relevant to infection management, and collectively identified opportunities for strengthening

community-based AMR education, prevention, and preparedness. Contributions included mapping current barriers to appropriate antimicrobial use, suggesting strategies for collaboration with traditional practitioners, and developing community-level preparedness measures such as outbreak reporting mechanisms, health education campaigns, and pathways for timely referral.

This participatory approach strengthened the voices of participants by drawing directly on their lived professional experience, particularly regarding the role of traditional medicine in community health and helped to balance knowledge asymmetries commonly found in global health research. Workshop materials were piloted to ensure linguistic accessibility and comprehension.

The researchers collected data through audio recordings, their own field notes, and written materials that participants produced during group work and submitted at the end of each workshop. The accuracy of the transcription process was validated through the use of field notes. Full transcripts the two workshops are attached to the paper as a supplementary document (S1 Data). This document represents the raw data used in the analysis and serves as supporting documentation to enhance transparency. The data were systematically analyzed using an inductive thematic content analysis, and key insights have been synthesized and presented in the results section, with illustrative participant quotes included to support transparency and reader comprehension. This is to enable readers to grasp the analysis without requiring direct interpretation of the raw transcript data.

The participants in the two workshops were distributed as outlined in Table 1 below.

The inclusion criteria encompassed health practitioners (medical doctors, nurses, and pharmacists) with an interest in AMR or who were actively working on AMR issues in practice. Gender, age, and disciplinary diversity was sought, with participants drawn from paediatrics, medicine, surgery, and pharmacy. Pharmacists were specifically included to strengthen discussions around drug prescription and dispensing practices impacting access to and use of antimicrobials. Exclusion criteria ruled out non-practising health practitioners, those unable to attend the majority of the workshops, those unwilling to participate, or those who had not practised for at least five years. Recruitment at the respective hospitals began on 30 June 2023 and concluded on 27 July 2023.

Thus, the workshops generated spaces for engagement and knowledge co-creation among health practitioners across medicine, nursing, and pharmacy [35–38]. The data analysis conducted by the authors was guided by the analytical framework detailed in the next section. The data analysis was conducted by the authors using an inductive thematic content analysis method [39], chosen for its theoretical flexibility in analysing qualitative data. Themes were not predefined as part of the analytical framework but emerged through repeated reading and re-reading of the workshop transcripts and notes. The analysis proceeded in several steps: first, reading through the transcriptions to gain an overview of the data, followed by in-depth re-reading to identify key sentences relevant to the research aim and questions. These sentences were then coded into meaningful words or phrases, which formed the basis for analytical categories and sub-categories.

**Table 1. Participating health practitioners across the two workshops.**

| Category | Females | Males | Total |
|---|---|---|---|
| *Parirenyatwa Hospital* | | | |
| Medical doctors | 4 | 6 | 10 |
| Nurses | 2 | 0 | 2 |
| Pharmacists | 1 | 0 | 1 |
| *Harare (Sally Mugabe) Hospital* | | | |
| Medical doctors | 3 | 5 | 8 |
| Nurses | 2 | 0 | 2 |
| Pharmacists | 2 | 0 | 2 |
| **Totals** | **14** | **11** | **25** |

Through iterative comparison and discussion, these categories were refined into three final themes that captured the most salient and recurring patterns across the data. Throughout the process, particular emphasis was placed on ensuring that participants' voices were made clear and carried forward into the presentation of results, with findings supported by direct quotations.

### Analytical approach

Prevention and preparedness are fundamental aspects of managing health challenges. Prevention focuses on reducing the incidence and impact of diseases before they occur, whereas preparedness involves planning and organizing resources to effectively respond to and manage health emergencies when they arise [4,19,40]. Both approaches are crucial for addressing health challenges, such as AMR, and improving community health outcomes. Implementing prevention measures, such as antimicrobial stewardship programs, under conditions of limited resources, poor infrastructure, and lack of healthcare access is a challenge. Without a preparedness component in AMR education, such conditions would leave countries less equipped to respond to resistant infections, leading to AMR outbreaks with an impact on health and wellness [14,41]. While prevention is a fundamental aspect of AMR education, a focus on prevention alone is insufficient to address the complexities of antimicrobial resistance. The inevitability of resistance, the need for proactive rather than reactive responses, the importance of adequate training for healthcare professionals, the risk of overreliance on stewardship programs, the lack of public engagement in preparedness, the vulnerabilities in LMICs, and the need for sustained R&D all highlight the critical limitations of a prevention-only approach [42–44]. To address AMR effectively, educational programs must integrate both prevention and preparedness to ensure a comprehensive and resilient response to this global health threat.

To this end, preparedness in AMR education encompasses developing ways of detecting, responding to, and managing AMR challenges when preventive efforts are insufficient. AMR preparedness includes medical practitioners equipping them with the knowledge and skills needed to address outbreaks of resistant pathogens, focusing on identifying cases of resistant infections, infection control, and what medicines to administer to resistant infections [45,46]. As noted above, preparedness in AMR education promotes proactive approaches to AMR, which is key for mitigating the consequences of AMR, particularly in LMICs such as Zimbabwe, with challenges related to healthcare resources and infrastructure [47,48]. Proactive approaches encourage anticipation of resistance, and thus enable different actors to develop contingency plans, including alternative treatments.

Furthermore, preparedness in AMR education builds healthcare system resilience that can, through coordination among diverse sectors and parts of society, withstand the pressures of resistance in case of outbreaks of resistant diseases, maintain community trust, and prevent the disruption of social and economic structures, often resulting from extensive resistance [12,49]. AMR education should thus promote interdisciplinary collaboration and sharing of best practices across these sectors [50,51]. Such resilient healthcare systems need to be nested within a resilient society, where preparedness cannot succeed without engagement with or involvement of communities. Research has highlighted that AMR education must be extended to community engagement to ensure that communities understand how they can contribute to reducing the spread of resistant infections [52,53]. Community engagement as part of developing preparedness through AMR education should include enabling community members to recognize signs of resistant infections, what to do in such a situation, who to turn to, and understand the necessity of engaging with medical practitioners and following their advice during outbreaks [20].

### Results

Three themes were inductively derived from the analysis of transcripts from participant discussions regarding the role of traditional medicine in addressing AMR in Zimbabwe. These include traditional medicine as: 1) *community healthcare*, especially in rural areas; 2) *complementary to conventional medical practices*; and 3) having a role in *AMR and disease*

*surveillance*. Together, these themes illustrate the main recurring patterns reflecting the discussions among the majority of participants across both workshops on how traditional medical practices can contribute to AMR education within ongoing healthcare practices.

## Community healthcare

Within many communities, particularly in rural areas, traditional medicine provides important healthcare, as access to conventional medicine and practitioners is limited. Because traditional medical practitioners often live and are part of the communities, traditional medical practices form the primary healthcare resource that is both accessible and familiar, creating trust in the communities in the treatments. This positioning, described by workshop participants, becomes relevant for AMR education as the traditional practitioner is the first point of contact for many patients and thus a potential avenue to include prevention and stewardship efforts into existing local practices. As highlighted in the following quote, this has an impact on addressing AMR.

> They are big on their own traditional treatments and it works for them. So I would say yes, and people in rural areas, there are different religions, tribes, you know, advocating and administering their medicines. (Participant 5, Parirenyatwa workshop)

> I think that if people are enlightened, as I was saying that in the rural areas, the communities, they support the traditional medicine. (Participant 1, Harare workshop)

> I also think traditional medicine has played a part in addressing AMR in the sense that maybe if you look at yesteryears, according to what we hear from our parents, it was normal prevention. You'll be given medication to stop you from getting this, you'll be given what you need to stop you from getting this. (Participant 1, Parirenyatwa workshop)

The participants highlighted how this medical practice prevents diseases that are common in the community. Furthermore, the following quote shows how traditional medical practitioners are part of the community.

> Again, because of [traditional practitioners] being closer to the potential patients, being of course more numerous generally, also being trusted in the communities. So they live within the communities and are likely to be one of the first to know if there are any new disease surfaces, for example, co-morbidities, TB. (Participant 2, Harare workshop)

> I think that the medical practitioners and those people who are doing the traditional medicine. I can see maybe the community, looking at the rural areas, communities are already on board. Because when people do integrated disease surveillance, they are being told maybe to look out for any animals that would be dying, maybe people who are experiencing certain disease and things like that, and they are reporting. (Participant 1, Parirenyatwa workshop.)

In participants discussions, traditional medicine was understood as community healthcare by being locally embedded and functioning as accessible first line of contact for patients, incorporating prevent common diseases, informal triage and early disease detection. Participants highlighted how living within the community means that traditional medical practitioners will be early in responding to health issues before the development of symptoms, which could be an entry point for community-based AMR awareness and early education initiatives. In addition, these quotes indicate that participants perceive traditional medicine as both more accessible and embedded within communities. Trust in local practitioners, coupled with their proximity to patients, was described by the participants as resulting in health concerns often being addressed earlier, and with the potential to embed AMR educational principles in everyday healthcare.

## Complementary with conventional medical practices

Traditional medical practices are often used by communities in Zimbabwean along with conventional medicine and can thus be considered complementary, which participants framed as pragmatic coexistence, with traditional medicine supporting immunity or symptom relief while conventional antibiotics addressed persistent infections This is highlighted by the participants in the following quote.

> My view is, really, there are plans at national level to integrate conventional and traditional, but practically, I feel like they are more on the complement side. They just coexist mainly. Of course, there are plans. Of course, we already have some councils, like you're saying, but at the moment, on the ground, for me, it's really some form of coexistence. (Participant 5, Harare workshop)

> I think like what Doc said, it's a great opportunity if we bring the two together, if we consider the traditional medicine as a preventive, let's say boost your immunity, and then you take your antibiotics. (Participant 2, Parirenyatwa workshop)

> Even during the COVID era, the Mzumbani traditional roots and beliefs became so popular that even the health workers, those who were practicing the conventional, were even convinced that this is working more, it's more effective than the conventional. So they used both. (Participant 3, Harare workshop)

> I think if we studied more, they could work hand in hand with the conventional tablets that we have already. Because it works to some. For some, they say it's gone. For some, it will come back after years. So we have to study more. That means we have to be open-minded even more to accept that there's something else other than just western medicines. (Participant 5, Parirenyatwa workshop)

As seen in the quotes, communities, including health workers, combined traditional herbal medicines with antimicrobials. This illustrates how traditional and conventional medicine come to coexist while the boundaries becoming blurred with associated AMR risks, such as ending antimicrobial courses too early or combining herbal remedies, as well as opportunities. Specifically, the quotes highlight an opportunity for AMR education of using points of intersections when treatments are combined to press home the importance of safe drug–herb interactions and responsible antimicrobial use. This is further emphasized by participants arguing that:

> I think actually the traditional practice is actually very important considering that most of our population, especially young people, they tend to vote for it first, mostly because that's what they can afford and that's what's available to them (Participant 4, Parirenyatwa workshop)

> I think traditional medicine, does offer an effect to the body, I guess, that will assist, let's say I might be taking antibiotics or anything whatsoever. If I'm more boosted and if I'm taking a certain part of traditional medicine, I feel like it will assist me to an extent whereby I might be able to fight that infection. (Participant 4, Harare workshop)

> I think addressing AMR in Zimbabwe, the traditional medicine has played a major role because when someone is prescribed to start taking their antibiotics, let's say they have a flu and their antibiotic is slow acting, it has to take them seven days, probably on day two they're not feeling any better. They might just stop taking their medicines and switch to the traditional medicines for a quick fix. So I feel like it has played a role, and to those that decide to take their full course and as well as take the traditional medicines, we really don't know the drug to help interactions that we have because we haven't really tested them out. We just know if you take your guava leaves and your lemon and what was, you mix them, you take them, they will help you ease your cough and everything, as well as you're taking your antibiotics. So maybe it will be added, we don't know. I can't say it was adding to it or maybe it was helping, but then it has some effect on AMR that we are having here in Zimbabwe. (Participant 5, Parirenyatwa workshop)

In the discussions, participants linked concerns about combining medicines and the risk of incomplete antibiotic courses, linking it to the risk of resistance. With traditional medicine often being the initial choice, including young people and communities with limited resources, integrating it with conventional medical practices can be framed as a arena for AMR education, connecting prevention and preparedness with responsible antimicrobial use. These reflections highlight the potential of conducting AMR education at intersection of traditional and conventional medical practices, including addressing the knowledge gap about correct antibiotic use. In bringing attention to these risks, participants highlighted areas of AMR education while appreciating role of traditional medical practices for communities.

This is illustrated in how the quotes also emphasise the value, according to the participants, of traditional medicine when used to support the health of patients taking antibiotics. Participants described how traditional medicine is often combined with conventional medical treatments. According to the health practitioners, traditional remedies were often initially used as symptoms emerged, and if there was no improvement, then pharmaceuticals such as antibiotics were sought.

**AMR and disease surveillance**

Participants emphasized how traditional medical practitioners play an important role in maintaining AMR and disease surveillance, which is understood as monitoring of resistance patterns and emerging and spreading disease symptoms to enable responses from the healthcare system. Participants highlighted that the visibility of traditional medical practitioners within communities positioned them to early identify symptoms and potentially contribute to outbreak reporting if communication channels with formal health systems were established. Rather than describing established formal reporting, results highlight participants perceived potential for engagements between medical practices with AMR education being key in realising this potential. This potential however is key to both effectively treating diseases and addressing AMR, as illustrated by the following quotes:

> So traditional medical practitioners can be completely effective in monitoring these outbreaks, and there's a very direct effect on them. They live often, they live in communities and are likely to be the first to work in any of these services. (Participant 2, Parirenyatwa workshop)

> The traditional medical practitioners could maybe be taught why, what, when, and how to report unusual symptoms in their patients to the health system. But of course this demands quite a lot of trust and a good relationship. So the suggestion of using, for example in the literature, the suggestion of using checklists and having these pictorial guides to symptoms, diseases, and modes of transmission, and facilitating communication between traditional medical practitioners and the community has potential. (Participant 2, Harare workshop)

> Open lines of communication between traditional medical practitioners and the community of conventional medical medicine could tremendously improve surveillance, That health officials must then include traditional medicine medical practitioners in the education outreach. (Participant 3, Parirenyatwa workshop)

The workshop participants noted that because they live within the community, traditional medical practitioners are aware of what is going on, as it is a direct concern of them both professionally and privately. With training, trust-building, and supportive tools such as symptom checklists or pictorial guides, participants suggested that traditional practitioners could be key in early reporting of symptoms to formal health systems before they worsen and spread. The perspectives raised in these quotes, illustrate how participants perceived AMR education not only as information transfer to communities but also as capacity-building for traditional practitioners to participate in early reporting and surveillance.

In addition, the quotes highlight how traditional medical practitioners, while not currently an integrated part of the formal surveillance systems have knowledge and relationships with communities to be part of AMR and disease surveillance on

the local level. Being present in the communities means that they are able to observe emerging health concerns, with participants pointing to the potential benefit of linking traditional medical practitioners to the more formal systems of disease surveillance and educational outreach.

While many of the participants' quotes do not explicitly reference AMR education, it was the focus of the workshops, and the quotes reflect relevant social and healthcare contexts for AMR education. Participants discussions were analysed as engaging with prevention and preparedness in addressing AMR, appropriate treatment practices, and the potential of integrating traditional medicine into AMR and disease surveillance, all themes indicating the key role of education in connecting traditional medicine with efforts to address AMR in Zimbabwe. Participants discussions of switching between treatments and the importance of trust and accessibility in for health seeking behaviours, highlights how knowledge, relationships and practices impact antimicrobial use and disease treatment. In their discussions, participants situated AMR education as part of these everyday practices and outlined a role for traditional medicine as part of opportunities to strengthen AMR education, responsible antimicrobial use, alternative treatments and building surveillance capacity. The results as a whole can thus be read as illustrating contextual conditions and points of engagement for developing AMR education in Zimbabwe, more than specific descriptions of existing AMR education initiatives.

## Discussion

In this section, we discuss how traditional medicine in Zimbabwe contributes to AMR preparedness, offering new perspectives on integrating traditional medical practices into AMR education. These results highlight the role of traditional medicine in a patient- and community-centric approach to AMR education, while also highlighting tensions and potential AMR risks related to the role of traditional medicine. The three themes presented in the results section emerged from an inductive analysis of participant workshop discussions. Together, they provide insights into how conventional health practitioners perceive the role of traditional medicine in AMR education and preparedness. Our results highlight how community trust and embeddedness are not limited to descriptive features of traditional medicine but constitutes structural conditions for preparedness with both opportunities and challenges. While trust enables early care-seeking and community adherence to health advice, embeddedness allow traditional medical practitioners to detect emerging symptoms before they get worse and spread, supporting surveillance against AMR. Meanwhile, trust for and reliance on traditional medicine might not be equally distributed throughout communities with this study not being able to sufficiently capture potential contradictory positions which could be the subject of further research. The study involved a relatively small number of participants, 25 across two workshops, which offers depth but limited generalisability of results across other contexts.

Results illustrate how traditional medicine offers accessible healthcare, particularly in rural areas, and is deeply embedded in the culture and social lives of Zimbabwean communities. This cultural rootedness is significant for AMR, as highlighted by Elton et al. [14] and Dyar et al. [41], forming a basis for strengthening preparedness efforts. Both the preventive care and early treatment offered through traditional medicine can reduce reliance on antimicrobials, thereby supporting broader AMR mitigation strategies. AMR education could benefit from integrating traditional medicine as a way to develop inclusive approaches that are relevant to communities' everyday lives and strengthen resilience. Traditional medicine remains a vital component of community healthcare, especially in areas with limited access to conventional services. Traditional practitioners, who are often part of their communities, provide care that is not only accessible but also culturally accepted and trusted. Crucially, workshop participants were all conventional medical practitioners, which meant that their training and professional background would have impacted how they perceive traditional medicine. In this healthcare context, trust becomes key for preventive practices to reduce reliance on antimicrobial treatments while also preparing communities for resistant infections outbreaks. This aligns with how [4] argues that challenges to set up AMR surveillance, especially in low- and middle- income countries (LMICs) include limited financial resources and technical capacity, emphasising how integrating traditional healthcare practices, which communities trust, could complement formal healthcare efforts while also requiring consideration and mitigation of associated risks. Furthermore [20], through a systemic review

highlighted how AMR awareness often involved incomplete knowledge and misconceptions. These knowledge gaps points to the importance of AMR education and how it involves drawing on both formal and traditional healthcare practices to strengthen public understanding of AMR. There are thus indications that traditional medicine can have multiple roles in addressing AMR, including being the first-stop healthcare for communities and providing alternatives to antimicrobials. Simultaneously, our results highlighted risks, including incomplete antimicrobial courses and combinations of conventional pharmaceuticals and traditional medicines, which could be areas for further AMR education. This preventive aspect of traditional medicine is crucial in the fight against AMR, as reducing the demand for antibiotics is a key strategy for slowing the development of resistance [4,5].

Traditional practitioners' proximity to and knowledge of their communities enable them to act as first responders during health crises. Their ability to intervene early positions them as key actors in strengthening preparedness against the spread of resistant infections. In the context of AMR, preparedness extends beyond outbreak response to include pro-active efforts to prevent and manage potential crises. As highlighted in the results, and consistent with Zhao et al. [28], traditional medicine's role in early disease detection and management contributes significantly to AMR preparedness by addressing health issues before they escalate. This study reveals that traditional medicine plays a vital role in community level preparedness, particularly through its contribution to early disease detection and management within the community. As noted by Mitchell et al. [26], traditional practitioners often serve as the first point of contact for emerging health issues, because of their integration within the community. Traditional practitioner's embeddedness in the communities allows for early detection and response to health threats, which is critical for effective AMR preparedness. These insights importantly cover primarily urban settings where the workshops were conducted, while the participants reported having experiences from rural health practices as part of their training. Furthermore, this potential is not realised as integration of traditional medicine in formal surveillance has not been done, reflecting ongoing tensions between capacities identified by partici-pating conventional health practitioners and existing institutional support for integrated AMR education. As these medical practitioners are typically the first to notice and address new disease trends, they are also key for community members to take part and shape health responses making them essential for community-level disease surveillance [12,14]. Traditional medical practitioners, who are often the first point of contact and remain continuously engaged within their communities during outbreaks, play an important role in shaping both short- and long-term antimicrobial use, thereby influencing efforts to address AMR. In addition to its practical value, such community-centric AMR educational approaches are transfor-mative in that they shift health decision-making closer to vulnerable groups affected by AMR promoting more equitable health. These benefits meanwhile depend on AMR education efforts being careful to address possible misuses of tradi-tional medicines, especially when combined with antimicrobials.

Moreover, the results highlight the complementary role of traditional medicine in supporting conventional treatments during outbreaks of drug-resistant infections. Results highlight how traditional remedies are often used alongside antibi-otics to manage symptoms and improve overall patient health. This complementarity represents opportunities as well as risks. Traditional medicine offers alternatives to antibiotics for managing less severe infections and acting as supportive therapy when used in combination with antibiotics. This could support patient recovery and reducing the strain on conven-tional healthcare resources, which is particularly valuable in resource-limited settings, such as Zimbabwe [17,27]. Mean-while, patients combining medicines raises questions about drug–herbal interactions and patient adherence that could contribute to AMR. These tensions highlight how traditional medicine harbours both potential opportunities and challenges for preparedness and AMR education, which would be valuable to explore in future research.

The results suggest that effective AMR preparedness requires active community engagement, with traditional medicine serving as a key facilitator. As trusted figures in their communities, traditional practitioners also contrib-ute to AMR education by raising awareness about antimicrobial misuse and the importance of timely, appropriate care [20]. Integrating traditional medicine into AMR education and preparedness strategies can empower communi-ties, as the trust placed in traditional practitioners strengthens community resilience to AMR challenges [14,42,44].

Meanwhile, the lack of more extensive contrasting views constitutes a limitation of this study with regards to understanding community possible scepticism towards traditional medicine. This paper argues that patient-centred, community-oriented approaches that involve traditional practitioners in disease surveillance and AMR education are essential for building a comprehensive and effective response to AMR [11,19]. To this end, future AMR education efforts could be developed to recognize traditional medicine not only as a healthcare resource but also as a contributor to greater health equity.

The integration of traditional medicine into AMR education, with an emphasis on preparedness, offers a multifaceted approach to managing antimicrobial resistance. Traditional medicine's role in community healthcare, its complementarity with conventional treatments, its contribution to disease surveillance, and the importance of scientific validation all strengthen the foundation of a robust AMR strategy. Realising this potential would demand addressing the risks and tensions identified in this study that are crucial to enable both safe and effective medical practice integration for AMR education. This paper demonstrates how traditional practices can enhance preparedness through community level AMR education and promote more proactive responses to resistance. Specifically, it contributes to AMR education by: 1) highlighting the role of traditional medical practices in community level AMR education and practice in Zimbabwe, and 2) broadening the field of AMR education and practice to encompass not only preventive action but also preparedness strategies grounded in patient- and community-centred approaches. By leveraging the strengths of traditional medicine, healthcare systems can improve their preparedness and response capacities, thereby contributing to more effective management of antimicrobial resistance.

A limitation of this study lies in its focus on health practitioners at two urban teaching hospitals. While participants had, as part of their training, experiences of traditional medical practices in the rural areas, further research could directly involve traditional medical practitioners and community members, especially to account for more contrasting and sceptical perceptions of traditional medicine. Future research could conduct similar workshops in more diverse contexts and engage traditional medical practitioners and community members. This would build on the results of this study to further explore community-centred perspectives of AMR education and provide deeper understanding s of tensions and challenges as part of Amr education.

This expanded discussion highlights the importance of incorporating traditional medicine into AMR education, emphasizing how this integration can lead to better health outcomes and more resilient healthcare systems. Incorporating preparedness into AMR education is not just beneficial; it is also essential. Given the global nature of AMR and its potential to disrupt healthcare systems, AMR education must adopt a holistic approach that encompasses both prevention and preparedness. Although prevention remains a critical element in addressing AMR, the complexities and unpredictability of resistance require a more robust and comprehensive approach. Integrating traditional knowledge and community leadership in AMR efforts as part of strengthening preparedness challenges top-down health approaches and promotes inclusion, while acknowledging the value of diverse health knowledges grounded in communities. In explicitly acknowledging, and through further research addressing, potential opportunities, challenges and risks in the role of traditional medicine enhances the relevance of these approaches. By integrating preparedness into AMR education, healthcare professionals and communities can be better equipped to handle the challenges of AMR, ensuring that healthcare systems remain resilient and capable of providing effective care. The inclusion of preparedness in AMR education is therefore a necessary step towards promoting community health and wellness and mitigating the impact of antimicrobial resistance on a local, national, and global scale.

## Supporting information

**S1 Data. Data set of workshop discussion transcripts from the two hospitals involved in the study.**
(PDF)

## Acknowledgments

We acknowledge the health practitioners (medical doctors, nurses, and pharmacists at Parirenyatwa University Hospital and Harare Central Hospital), who contributed greatly to the research through their participation in the participatory research workshops.

## Author contributions

**Conceptualization:** Martin Mickelsson, Tungamirirai Simbini.

**Data curation:** Martin Mickelsson.

**Formal analysis:** Martin Mickelsson.

**Funding acquisition:** Martin Mickelsson.

**Investigation:** Martin Mickelsson.

**Methodology:** Martin Mickelsson, Tungamirirai Simbini.

**Project administration:** Martin Mickelsson.

**Resources:** Martin Mickelsson.

**Software:** Martin Mickelsson.

**Validation:** Martin Mickelsson, Tungamirirai Simbini.

**Visualization:** Martin Mickelsson.

**Writing – original draft:** Martin Mickelsson.

**Writing – review & editing:** Martin Mickelsson, Tungamirirai Simbini.

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
