## [Decision Letter · Decision Letter 0]

8 Jul 2025

PGPH-D-25-01326

Integrating Traditional Medicine into Antimicrobial Resistance Education: Community-Centric Preparedness in Zimbabwe

Dear Dr. Martin Mickelsson,

Thank you for submitting your manuscript to PLOS Global Public Health. After careful consideration, we feel that it has merit but does not fully meet PLOS Global Public Health’s publication criteria as it currently stands. Therefore, we invite you to submit a revised version of the manuscript that addresses the points raised during the review process.

We look forward to receiving your revised manuscript.

Kind regards,

Muhammad Asaduzzaman, MD MPH MPhil

Academic Editor

Journal Requirements:

1. Please provide additional details regarding participant consent. In the ethics statement in the Methods and online submission information, please ensure that you have specified (1) whether consent was informed and (2) what type you obtained (for instance, written or verbal, and if verbal, how it was documented and witnessed).

2. Please include a complete copy of PLOS’ questionnaire on inclusivity in global research in your revised manuscript. Our policy for research in this area aims to improve transparency in the reporting of research performed outside of researchers’ own country or community. The policy applies to researchers who have travelled to a different country to conduct research, research with Indigenous populations or their lands, and research on cultural artefacts. The questionnaire can also be requested at the journal’s discretion for any other submissions, even if these conditions are not met. Please find more information on the policy and a link to download a blank copy of the questionnaire here: https://journals.plos.org/globalpublichealth/s/best-practices-in-research-reporting. Please upload a completed version of your questionnaire as Supporting Information when you resubmit your manuscript.

3. Please amend your detailed Financial Disclosure statement. This is published with the article. It must therefore be completed in full sentences and contain the exact wording you wish to be published.

i. Please clarify all sources of funding (financial or material support) for your study. List the grants (with grant number) or organizations (with url) that supported your study, including funding received from your institution. 

ii. State the initials, alongside each funding source, of each author to receive each grant.

iii. State what role the funders took in the study. If the funders had no role in your study, please state: “The funders had no role in study design, data collection and analysis, decision to publish, or preparation of the manuscript.”

iv. If any authors received a salary from any of your funders, please state which authors and which funders.

4. We have noticed that you have uploaded Supporting Information files, but you have not included a list of legends. Please add a full list of legends for your Supporting Information files after the references list.

5. In the online submission form, you indicated that “Anonymized data presented in this study are available on reasonable request from the corresponding author.”. 

3. Uploaded as supplementary information.

Additional Editor Comments (if provided):

Reviewers' comments:

Reviewer's Responses to Questions

**Comments to the Author**

1. Does this manuscript meet PLOS Global Public Health’s publication criteria ? Is the manuscript technically sound, and do the data support the conclusions? The manuscript must describe methodologically and ethically rigorous research with conclusions that are appropriately drawn based on the data presented.

Reviewer #1: Yes

Reviewer #2: No

2. Has the statistical analysis been performed appropriately and rigorously?

Reviewer #1: N/A

Reviewer #2: No

3. Have the authors made all data underlying the findings in their manuscript fully available (please refer to the Data Availability Statement at the start of the manuscript PDF file)?

Reviewer #1: Yes

Reviewer #2: No

4. Is the manuscript presented in an intelligible fashion and written in standard English?

Reviewer #1: Yes

Reviewer #2: No

5. Review Comments to the Author

Reviewer #1: The study explores how traditional medicine (TM) can be integrated into antimicrobial resistance (AMR) education to enhance preparedness efforts in Zimbabwe. Recognizing that AMR education has focused heavily on prevention, the paper argues for a shift toward preparedness, especially within low- and middle-income countries (LMICs), where healthcare resources are limited.

Methods

The method is adequate and concise. The authors used participatory research workshops and audio-recorded discussions to assess AMR preparedness according to the thematic roles.

Results and discussion

Well-written and concise

Conclusions:

Well-written

The paper calls for AMR policies and educational programs to integrate traditional medicine, arguing that such inclusion promotes community trust, health equity, and system resilience. It advocates for a dual prevention-preparedness model, especially critical in LMICs where AMR poses a major health threat.

The study makes a valuable contribution to global health education by challenging top-down approaches and promoting inclusive, culturally grounded strategies for AMR preparedness.

Reviewer #2: OVERALL COMMENT: If the manuscript can show how traditional medicine practically and educationally intersects with AMR, it has the potential to make a significant contribution. It contains important ideas. Before it is ready for publication, though, it needs to be significantly restructured, nearly completely rewritten, and subjected to more data analysis, presentation, and deeper engagement with both data and conceptual framing.

- Introduction—Requires Major Revision—Rewrite

* Meant for the entire introduction:

Comment 1: Use an active voice overall.

Comment 2: The authors make meaningful arguments, and there is a clear focus on equity and integration of traditional medicine in AMR preparedness—which is an interesting perspective.

However, I would suggest a rewrite of the introduction, as I have identified several areas for improvement.

1. The introduction jumps between prevention and preparedness without clearly demarcating each point or their interconnectedness. I would restructure the introduction more clearly.

2. Logical Consistency and Accuracy—Highlighted below on how microbes develop resistance.

3. Repetitiveness: Points regarding prevention’s insufficiency, risks, and vulnerability of communities are repeated multiple times without sufficient elaboration or added nuance.

4. Justification for Vulnerable Communities: The exact reason why these communities would be disproportionately affected by prevention-only approaches is not clear.

5. The link between traditional medicine and preparedness is promising but not sufficiently justified or explained.

* P3L11-12: The transition to Zimbabwe's systemic vulnerabilities and AMR education "vulnerabilities" is not logical. The previous sentence talks about education, while the latter sets up the system in Zimbabwe; just because there is imbalance doesn't make the analogy stand. I would make Zimbabwe's situation clearer.

The following sentence, L15, makes the assumption that the inequalities in the overall system are due to educational imbalance. Although this might be the point the authors are trying to make, the way they are making it doesn't hold up to me; I would need more depth.

*P3L13: exacerbate

*P3L21–P4L14: I would suggest a rewriting of the paragraph. Where points are made clearer. The authors shift between why prevention is not enough and that it induces risks, then talk about what prevention is and what could happen when prevention does not work. The purpose of this section to me is to convince the reader that prevention is not enough, and that does not translate here. I found myself scrolling up multiple times and trying to decipher the info shared to serve the point trying to come across.

*P3L21-P4L3: This seems to be the real reason. I think it should be placed higher within the manuscript; it emphasizes the importance of preparedness in AMR education better and sets the stage better.

*P4L6: A standalone sentence does not look good; the reasoning behind AMR education includes not transferring well. Reasoning needs to be structured more clearly.

* P4L8-L10: The flow and logic of this sentence are off. It sounds like microbes develop resistance to obtaining the mutations and horizontal gene transfer, whereas in reality these are the ways that they gain resistance; the current wording is unclear and does not convey this message.

*P4L12: Should be surveillance.

*P4L12-L14: Would need more detail on how a blind spot is created.

*P4L15-L20: I fail to understand why prevention-focused AMR education limits the capacity to deal with resistant infections, as to my point for the paragraph ending with P4L14.

*P4L20: limitations

*P4L20-21: This point has been made multiple times but lacks the structure to understand. I understand the point that prevention-only education increases risks, but why the vulnerable communities are more affected is not clear to me.

*P22-P25: "Lack of sufficient preparedness, medical practitioners, and healthcare services could stand without the tools..." is unclear and awkwardly phrased. Consider rewriting as

"Without sufficient preparedness, medical practitioners and healthcare services may lack the tools, protocols, and resources..."

*P5L4-L9: The transition is a bit rough and abrupt. A clear bridging sentence would work nicely while narrowing the focus.

*P5L15: Weird way of citing. I would get rid of it.

*P6L3-L5: Needs a logical follow-through on why these questions were formulated. For instance, is it because there are gaps in traditional medicine's role both globally and in Zimbabwe? This should be clearer.

I would emphasize the novelty and necessity of the gap your research addresses. This also refers to the lack of a clear justification of research focus.

*****

- Materials and Methods —Requires Major Revision - Rewrite and add content

* Meant for the entire Materials and Methods:

Comment: The section has foundational content but requires clearer presentation and improved structure. Below are general points the authors should address:

Clearly outline the exact structure and key activities of the workshops, even if briefly. Simply stating that participatory workshops were conducted is insufficient without describing what was done during them.

Use tables or lists to improve readability, especially when reporting participant characteristics.

Justify methodological choices explicitly, including why participatory workshops were appropriate in this context and how they build on the cited literature.

The section is overly repetitive, and sentence structure is often needlessly complex. Revise for conciseness and clarity.

Conceptual explanations should be placed in the Introduction or Discussion. The Methods section should only describe what was done to generate the data presented in this study.

Clearly articulate and simplify the analytical approach. Avoid theoretical elaboration; just state the framework and how it was applied to the data.

Improve the overall organization of the section to ensure a logical and consistent flow.

*P6L9–L11:

While the authors refer to previous work, a brief summary of the original method would be helpful in understanding how the current study builds upon or adapts that approach.

*P6L11–L17:

The authors describe the outcomes of the participatory workshops, but not the activities themselves. Please mention what concrete tasks or exercises were conducted with participants (e.g., discussion prompts, group mapping, scenario building). Link outcomes directly to the activities used to generate them.

*P6L17–19:

This content overlaps with the earlier lines (P6L9–L11) and could be consolidated to reduce repetition.

*P6L24–P7L4:

The description of participant characteristics is difficult to follow in paragraph form. I strongly recommend using a table to present participant roles, numbers, gender, and affiliation by hospital for improved readability.

*P6L6–L11:

It would be helpful to include statistics or descriptors indicating participants’ professional engagement levels with AMR and AMR education, similar to the way their job roles and gender are reported.

*P6L11:

There is a shift from passive to active voice (“we”) that breaks consistency. I recommend aligning voice throughout the section—either consistently active or passive—to improve readability.

*P6L15–16:

It is unclear whether recruitment at both hospitals began simultaneously, or if the dates listed correspond to only one of them. Please clarify this.

*P6L20–L22:

This sentence again states that methods were adapted from a previous study but lacks detail. If the outcome of the method is described, the authors should briefly mention how the adaptation was made or implemented in practice.

*P6L23:

Please indicate how transcription quality was assured. For instance, was it cross-checked, reviewed by multiple researchers, or validated using field notes?

*P6L24:

“Contemporary field notes” is unclear. Consider rephrasing to simply “field notes taken during the workshops.”

*P7L3:

The entire “Analytical Approach” section reads like an extension of the Introduction or Discussion. The Methods section should focus on describing the analytical framework used, not reiterating why prevention vs. preparedness matters. A more concise and technical explanation is needed (e.g., thematic coding, grounded theory, etc.).

*P10L19–20:

This sentence is redundant with P10L17–18 and should be removed or merged for clarity.

*****

- Results—Major Revision - Rewrite and add content

*Meant for entire results:

The Results section raises several concerns related to data presentation, analytical depth, and alignment with the study’s AMR focus. While the topic is important, the current presentation lacks the clarity, rigor, and evidentiary support expected from qualitative research. Below are key areas requiring revision:

1. Insufficient Use of Data to Substantiate Claims

Across all themes, the authors rely on a limited number of participant quotes (2–3 per theme) from a sample of 25. This raises concerns about the representativeness and saturation of the data. The findings are presented as broadly generalizable insights, yet there is no information on 1) how many participants contributed to each theme, 2) whether these views were shared across both workshops, or 3) the presence of any dissenting or contrasting views. This creates the appearance of anecdotal reporting rather than analytical saturation.

Action:

Specify how many participants contributed to each theme and whether these views were shared across sites. Report whether the themes reflect majority, minority, or contested positions. Consider using descriptive statistics or visualizations (e.g., quote frequency tables) to improve transparency and interpretability.

2. Shallow Thematic Development

Thematic categories are introduced but not critically unpacked. Core ideas (e.g., trust, accessibility, community embeddedness) are repeated without deeper analysis or qualification. In addition, it remains unclear how the three themes were derived: were they predefined, emergent, or adapted from previous work?

Action:

Clarify the process of thematic derivation—this relates to both Methods and Results. State explicitly whether the themes were developed inductively or deductively, and support them with a broader range of quotes and participant input to avoid superficial treatment.

3. Terminology Ambiguity

Key terms such as “community healthcare”, “surveillance”, and “complementarity” are used without definition. For example, “community healthcare” is central to the first theme but remains vague—does it refer to first-line care, traditional prevention, or informal triage?

Action:

Introduce brief operational definitions for key terms at the start of the Results and use them consistently.

4. Overinterpretation and Causal Claims Without Evidence

Several claims go beyond what the presented data support—for instance, statements about traditional medicine reducing antimicrobial use or strengthening health system resilience. Quotes provided do not demonstrate behavioral shifts, systemic integration, or measurable AMR outcomes.

Action:

Restrict the results to descriptive findings. Move interpretive or causal claims (e.g., AMR impact, health system strain) to the Discussion, where they can be explored with appropriate qualification.

5. Underdeveloped Link to AMR Education

Although AMR education is the central stated focus, most results remain tangential to this theme. The participant quotes rarely reference AMR education, antimicrobial use, or knowledge-building directly. This weakens the connection between the data and the study’s stated contribution.

Action:

For each theme, explicitly connect findings to AMR education—either through participant quotes, or clear synthesis by the authors. Show how traditional medicine influences or intersects with antimicrobial literacy, stewardship practices, or community-based preparedness education.

6. Quote Integration and Transcription Issues

Some quotes are grammatically unclear or appear paraphrased. Attribution formatting is inconsistent, and one quote appears to contain a broken or incomplete sentence.

Action:

Ensure all quotes are transcribed accurately and formatted consistently (e.g., with participant ID and workshop site). If quotes are paraphrased or translated, clarify this in the Methods.

Minor Issues:

Redundancy: Several points—particularly community trust, practitioner embeddedness, and access—are repeated across sections without further analytical elaboration.

Speculative phrasing: Avoid phrases like “this shows that…”, “this may reduce demand…” unless these are directly supported by data.

Terminology clarity: Terms like “preparedness” and “surveillance” must be anchored in context and clearly distinguished from their use in formal health systems.

*P10L22–23: This is already a given. No need to repeat.

*P10L24–P11L1: Language is interpretive. Avoid drawing conclusions before data is presented.

*P11L3–L4: This is interpretive and assumptive. The themes are stated to be identified, but no supporting evidence has been provided yet.

*P11L6–L10: Phrases like “provides important healthcare” are speculative. If the section is focused on workshop findings, results should reflect participant quotes only—avoid editorializing. Reframe these as descriptive summaries grounded in data.

*P11L10–L11: Sentence beginning with “As highlighted…” is speculative. A more neutral transition, such as “One participant reflected on…,” would improve clarity and tone.

*P11L15: “Referring to ”this”—expand or clarify the quote for better context and readability.

*P11L17–18: This is speculative and largely repeats the quote. Rephrase neutrally or remove.

*P11L18–P12L1: This functions as a meta-statement and reduces coherence. Rephrase or integrate more smoothly.

*P12L1:

The phrase “will be early” is overly casual in tone—rephrase for neutrality.

*P12L2–4:

Highly speculative. The suggestion that two quotes support antimicrobial use reduction is unsubstantiated. This content belongs in the Discussion if adequately supported—otherwise, remove it.

*P12L7:

The term “complementary” is underdefined. Clarify whether complementarity refers to behavior, improved adherence, delayed access, or clinical outcomes. At present, the usage lacks analytical depth.

*P12L8:

The sentence beginning with “This is highlighted…” is unnecessary and redundant.

*P12L13–14:

Redundant phrasing—streamline.

*P13L1–L3:

The claim that traditional medicine offers alternatives to antibiotics is not supported by the data. Participants describe simultaneous use, not substitution. Remove or revise.

*P13L3–L6:

This is a major extrapolation. Nowhere in the data is it stated that traditional medicine reduces prescriptions, changes behavior, or impacts resistance. Move this to the discussion and qualify it appropriately, or remove it.

*P13L7 (entire section):

This section does not align with the stated scope of the study (AMR education). If retained, the authors must 1) define what “surveillance” means in this context; 2) provide evidence of actual engagement or referral behavior by traditional practitioners; and 3) clearly connect findings to AMR education outcomes.

*P13L8:

“Play a crucial role” is too strong a claim given the limited and vague supporting evidence. Revise language for accuracy.

*P13L9:

The term “AMR and disease surveillance” needs definition. Quotes imply a loose relationship but provide no evidence of how traditional practitioners engage with formal surveillance efforts.

*P13L11–L14:

The quote is hypothetical (“can be”), not empirical. Authors must acknowledge the speculative nature or provide evidence that traditional practitioners are involved in AMR surveillance.

*P13L16–L19:

This quote is grammatically broken and unclear. Either correct, exclude, or summarize as paraphrased (and state that explicitly).

*P13L23–L25:

The claim that “traditional medical practitioners play a crucial role…” is a strong overstatement not supported by the data. Rephrase to reflect participant perspectives more accurately or remove.

******

Discussion:

The Discussion attempts to reframe AMR education through a community-centric lens by highlighting the potential role of traditional medicine in prevention, surveillance, and preparedness. While the ambition is commendable, the Discussion suffers from:

1. Overinterpretation of limited data:

-- The data presented in the Results section of the manuscript are not strong enough to justify claims made in the discussion section. The manuscript uses 25 participants and uses only 2-3 per theme.

-- Temper all such claims (such as community surveillance being driven by traditional practitioners or traditional medicine being "key" for AMR surveillance) with conditional or speculative language within the discussion.

2. Redundancy and verbosity: The discussion would benefit from a clearer structure with content blocks for superior readability and focus, where each paragraph should only make a single well-supported point.

3. Conceptual overreach: Define key terms clearly.

4. Uncritical literature integration: The use of citations at P14L16 and L22 does not support a claim. Mention why the citations are relevant; explicitly use citations only to support the claim and position made.

5. Lack of engagement with limitations and nuance:

-- The only limitation is set to be that health practitioners were included.

- However, there is no reflection on:

-- The low number of people 25 with only two workshops and the strength of the claims being made.

-- Potential bias of formally trained personnel on traditional medicine.

-- Urban settings skewing perceptions about traditional practices.

6. Neglect of Contradictions or Tensions: A robust discussion should acknowledge complexity, not just convergence. No mention of possible skepticism towards traditional medicine is mentioned; any divergent views were also not shared. Even if they do not exist, the authors should explicitly state this is a limitation.

*****

- Other -

*P18L22: Isn't this PLOS One? I suspect the authors have already submitted the manuscript to another journal. Please make sure the manuscript is submitted in the correct format.

*Regarding data availability: The authors have provided a 52-page document, which I assume to be the full transcript of the two workshops; however, this is neither explicitly stated nor referenced within the manuscript. The nature and purpose of the supplementary data should be clearly described. More importantly, the data—if intended as a transcript—needs to be processed and presented in a way that supports reader comprehension. In its current form, the raw data is not meaningfully integrated into the analysis. The authors are expected to analyze the data and reflect key insights within the Results section, rather than relying on readers to extract meaning independently.

6. PLOS authors have the option to publish the peer review history of their article (what does this mean? ). If published, this will include your full peer review and any attached files.

**Do you want your identity to be public for this peer review?** For information about this choice, including consent withdrawal, please see our Privacy Policy .

Reviewer #1: No

Reviewer #2: **Yes: ** Ege Dedeoğlu

---

## [Decision Letter · Decision Letter 1]

9 Oct 2025

Integrating Traditional Medicine into Antimicrobial Resistance Education: Community-Centric Preparedness in Zimbabwe

PGPH-D-25-01326R1

Dear Martin Mickelsson,

We are pleased to inform you that your manuscript 'Integrating Traditional Medicine into Antimicrobial Resistance Education: Community-Centric Preparedness in Zimbabwe' has been provisionally accepted for publication in PLOS Global Public Health.

Best regards,

Muhammad Asaduzzaman, MD MPH MPhil

Academic Editor

Reviewer Comments (if any, and for reference):

Reviewer's Responses to Questions

**Comments to the Author**

1. If the authors have adequately addressed your comments raised in a previous round of review and you feel that this manuscript is now acceptable for publication, you may indicate that here to bypass the “Comments to the Author” section, enter your conflict of interest statement in the “Confidential to Editor” section, and submit your "Accept" recommendation.

Reviewer #2: All comments have been addressed

2. Does this manuscript meet PLOS Global Public Health’s publication criteria ? Is the manuscript technically sound, and do the data support the conclusions? The manuscript must describe methodologically and ethically rigorous research with conclusions that are appropriately drawn based on the data presented.

Reviewer #2: Yes

3. Has the statistical analysis been performed appropriately and rigorously?

Reviewer #2: Yes

4. Have the authors made all data underlying the findings in their manuscript fully available (please refer to the Data Availability Statement at the start of the manuscript PDF file)?

Reviewer #2: Yes

5. Is the manuscript presented in an intelligible fashion and written in standard English?

Reviewer #2: Yes

6. Review Comments to the Author

Reviewer #2: (No Response)

7. PLOS authors have the option to publish the peer review history of their article (what does this mean? ). If published, this will include your full peer review and any attached files.

**Do you want your identity to be public for this peer review?** For information about this choice, including consent withdrawal, please see our Privacy Policy .

Reviewer #2: **Yes: ** Ege Dedeoğlu
